# Tobramycin Blood Levels after Local Antibiotic Treatment of Bone and Soft Tissue Infection

**DOI:** 10.3390/antibiotics11030336

**Published:** 2022-03-04

**Authors:** Carlos D. Pargas, Ahmed H. Elhessy, Mehdi Abouei, Martin G. Gesheff, Janet D. Conway

**Affiliations:** International Center for Limb Lengthening, Rubin Institute for Advanced Orthopedics, Sinai Hospital of Baltimore, Baltimore, MD 21215, USA; ccpargas@gmail.com (C.D.P.); ahmed_hessy@hotmail.com (A.H.E.); mehdi5675@yahoo.com (M.A.); mgesheff@lifebridgehealth.org (M.G.G.)

**Keywords:** acute kidney injury, calcium sulfate, creatinine, PMMA, safety, tobramycin, toxicity

## Abstract

Local antibiotic delivery using different carriers plays an important role in both infection prophylaxis and treatment. Besides dead space management, these carriers have the advantage of providing a high concentration of local antibiotics with a lower risk of systemic toxicity. Few studies have reported on systemic toxicity associated with antibiotic-impregnated carriers. The present study investigates the systemic tobramycin concentration at 24, 48 and 72 h postoperatively after using tobramycin-loaded polymethyl methacrylate (PMMA) and calcium sulfate (CS) as local antibiotic carriers. Additionally, this work assesses the renal function postoperatively for indications of acute kidney injury (AKI). Fifty-two patients were treated in 58 procedures with tobramycin and vancomycin-loaded PMMA, CS, or both. All systemic tobramycin levels were <2 mcg/mL at 72 h, and the resulting rate of AKI was 12% (7/58). In conclusion, local tobramycin antibiotic delivery using PMMA, CS, or both remains a safe and effective modality in the treatment of osteomyelitis as long as the surgeon is aware of its possible nephrotoxic effect.

## 1. Introduction

Postoperative infection is a potentially catastrophic complication in orthopedic pa-tients, and its management remains challenging [1]. Successful management usually involves multiple treatment modalities including surgical debridement of the infected tissue, removal of implants, dead space management, and a combination of systemic and local antibiotic delivery [2,3].

Combining surgical debridement with local and systemic antibiotics has been shown to be the most effective in the treatment of osteomyelitis [4]. Local antibiotics carry the advantage of delivering high local antibiotic concentration without toxic systemic levels. Various delivery systems are available for antibiotic delivery. Two of the antibiotic carriers that have been commonly used are polymethyl methacrylate (PMMA) and calcium sulfate (CS).

PMMA emerged as a very effective antibiotic carrier of choice in the 1970s [5], with its non-resorbable nature which can provide structural support and fill the dead space associated with larger bone defects. It is very effective in antibiotic delivery, as demonstrated by the 2017 Karek et al. [6] review of PMMA antibiotic elution using chest tube-coated guide rods and locked nails. At least 50% of the antibiotics eluded in the first 24 h, and the remaining antibiotic elution was at a sufficient level to inhibit *Staphylococcus aureus* for 6 weeks [6]. There are disadvantages with PMMA, however, including its presence as a foreign body which necessitates removal and subsequently filling the void with bone graft or other live tissue, as well as incomplete and inconsistent release of antibiotics [2]. CS usage has been popularized within the past decade as a modern antibiotic carrier with the advantages of drug-elution predictability, timed resorption, and the potential for host bone to gradually replace and fill the void, although it does not provide immediate structural stability [2,3,7,8].

Aminoglycosides—specifically tobramycin in combination with vancomycin delivered locally—create a synergistic effect against bacteria, which is favored in the management of osteomyelitis [9,10]. Systemic antibiotics, particularly tobramycin, have many adverse side effects with prolonged use and may cause kidney injury, but the use of antibiotic-loaded carriers achieves bactericidal concentrations locally without systemic toxicity [11]. One of the only disadvantages to local antibiotic carriers is that the systemic absorption is not entirely predictable [12].

Studies have also shown that systemic exposure of tobramycin loaded in CS appears to remain in safe margins (≤2 mcg/mL after 5 days post-implantation), except in patients with diminished renal function, which necessitates additional caution [2,3]. In 2005, Swieringa and Tulp reported toxic serum levels of gentamicin in 7 of 12 patients treated with gentamicin-loaded sponges, and 6 of them had a drop in renal clearance, although 3 of these proved to be temporary [13]. Several studies on PMMA as an antibiotic carrier using tobramycin or vancomycin have demonstrated a surge in systemic antibiotics elution only in the first few hours after surgery, followed by a rapid descent in blood levels over the next 24–48 h [14,15,16]. Systemic toxicity, however, does occur with antibiotic laden PMMA as well and there have been several case studies where locally delivered tobramycin or vancomycin has been reported to result in acute kidney injury (AKI) [17,18,19,20].

There are few studies associating the local concentrations of antibiotics and their resulting systemic blood concentrations with a combination of carriers [18,21]. There is additionally a scarcity of large studies documenting local antibiotic dosing and resulting postoperative kidney injury [21]. In this present study, 52 patients who underwent 58 procedures were investigated for the systemic tobramycin concentrations occurring at 24, 48 and 72 h postoperatively following treatment of orthopedic infections with a combination of local vancomycin/tobramycin laden antibiotic PMMA spacers, CS spacers, and combinations thereof. Preoperative and postoperative renal function was evaluated for any AKI as a direct result of the local antibiotic regimen.

## 2. Results

Our study cohort of 52 patients underwent 58 procedures and had a mean age at surgery of 61.7 ± 12.0 years (range, 19.3–85.0 years) with 36 (62.1%) being male. Mean BMI at the time of surgery was 33.2 ± 8.8 kg/m^2^ (range, 19.9–53.1 kg/m^2^) and mean body weight at time of surgery was 96.9 ± 28.8 kg (range 47–156 kg) (Table 1).

Table 1 lists the demographics, medical comorbidities, and baseline creatinine in our cohort. There were 8 (13.8%) cases with chronic kidney disease, 39 (67%) cases used nephrotoxic medications, and 23 (39.7%) cases had diabetes mellitus at the time of procedure. The average baseline creatinine in this population was 1.0 ± 1.0 mg/dL. Out of the 39 patients who were on nephrotoxic medications preoperatively, 41% (*n =* 16) were on statins, 30.8% (*n =* 12) were on angiotensin-converting enzyme (ACE) inhibitors, 20.5% (*n =* 8) were on vancomycin, and 17.9% (*n =* 7) were on angiotensin II receptor blockers (ARBs).

Regarding indications for surgery, periprosthetic knee infection was the most common, occurring in 56.9% (*n =* 33) of procedures. The other notable indications were hip infection (20.7%, *n* = 12) and osteomyelitis of tibia or femur (13.8%, *n* = 9). Of these 9 cases with fracture related infection (FRI), 7 had infection with retained implants and 2 had persistent infection after implant removal [22]. There was 1 (1.7%) lumbosacral wound and 3 cases of prophylactic insertion (2 total knee arthroplasties (3.4%) and 1 hip heterotopic bone excision (1.7%)) (Table 1).

Table 2 chronicles the details of the antibiotics, their carriers, and their administration. Ten cases received only CS with a mean dose of 14.0 mL (range, 10–20 mL) for a mean tobramycin dose of 1.7 g (range, 0.6–4.2 g). Six patients had 10 cc packs and 4 had 20 cc packs. Ten cases received only PMMA spacers with a mean of 5 units (range, 3–10) for a mean tobramycin dose of 8.8 g (range, 3.6–20.8 g). Thirty-eight cases received a combination of CS and PMMA. Within the combination cases, the mean dose of PMMA was 5 units (range, 3–8) and the mean dose of CS was 15.8 mL (range, 10–40 mL). Eighteen combination patients received 10 cc packs, 19 received 20 cc packs, and 1 received a 40 cc pack. The mean tobramycin dose in these combination cases was 9.7 g (range, 2.4–19.2 g). Of the total group, 7 patients did not have well documented tobramycin dosage, and these were excluded from the descriptive statistics and correlation analysis.

Thirty cases had tobramycin serum blood level measurements at all 3 intervals: 24, 48, and 72 h. Mean detectable tobramycin blood level at 24 h was 1.60 mcg/mL (range, 0.34–9.45 mcg/mL) (Figure 1). Systemic toxicity levels were considered to be when the serum level was higher than 8 mcg/mL. Subsequently, the level declined and at 48 h the average was 0.62 mcg/mL (range, 0.29–4.15 mcg/mL) (Figure 2). Finally, mean tobramycin blood level at 72 h was 0.41 mcg/mL (range, 0.29–1.95 mcg/mL) (Figure 3). Mean preoperative creatinine level was 1.04 mg/dL (range, 0.44–7.78, ±SD 1.04 mg/dL).

Tobramycin blood levels were collected at 24, 48, and 72 h postoperatively (*n* = 42, *n* = 35, and *n* = 55, respectively; Table 3). At 72 h postoperatively, all cases had tobramycin levels measuring inferior to toxicity levels with a mean result of 0.38 ± 0.32 mcg/mL (range, 0.29–1.85 mcg/mL). The 72 h tobramycin measurement was not obtained for 5 patients within the cohort; however, at 48 h all 5 of them reported <0.30 mcg/mL. At 72 h, 43 cases (78%) had undetectable levels, 2 cases (4%) recorded levels of 1.83 and 1.95 mcg/mL, and the remainder of this cohort (10 cases, 28%) all had levels < 1.0 mcg/mL (Table 3).

Only 3 cases had tobramycin levels > 5.0 mcg/mL and these all occurred at the 24 h interval (Table 3). One case had a level of 9.45 mcg/mL with the toxic cut-off, as per our lab, of >8.0 mcg/mL at the 24 h timepoint. These 3 cases all resolved by 72 h to <2 mcg/mL. Of the spiking cases, the initial tobramycin dose was >12 g in all instances. There was 1 AKI following this dose of tobramycin; that case had 19 g. The AKI resolved within 1 week. There was a correlation between postoperative tobramycin levels and grams of tobramycin used in all 3 groups with a correlation coefficient 0.55, 0.63, and 0.44 for the 24, 48, and 72 h timepoints, respectively.

There were 7 cases with stage 1 AKI, with 5 of these having an increase in creatinine of 0.3 mg/dL within 48 h, and the remaining 2 had a creatinine ≥ 1.5 times the baseline (Table 4). Of these AKI patients, 2 had history of chronic kidney insufficiency (CKI) preoperatively (GFR < 60). The doses of tobramycin in all these cases ranged from 2.4 g to 19e g with a mean of 9.6 g (Table 5). All AKIs resolved back to baseline creatinine within 1 week except for 1 patient who went on to CKI (peak Cr 3.29) but did not require dialysis. This patient did not have a CKI preoperatively and the creatinine gradually returned to baseline over the course of 8 months. No dialysis was necessary in any of the AKI patients. The amount of AKI cases in this study (12%) is comparable to other published series. The number of AKIs was too small to perform a regression analysis for risk factors in this group predisposing these patients to AKI. Of the 52 patients, 46 were available for follow-up at 1 year, with 87% of infections eradicated.

## 3. Discussion

The battle against infections in orthopedic surgery has always been challenging, and carriers used for the local delivery of antibiotics have become a formidable weapon in this fight [7]. Their chief drawback is renal problems; AKIs resulting from high-dose local antibiotic carriers have been reported as high as 14% in nonrenal patients and as high as 45% in patients with CKI [21]. Dagenaux et al. recently documented AKI in 455 cases of periprosthetic joint infections with an overall rate of 19% [21]. They reported on a variety of dosing regimens using only Simplex PMMA (Stryker, Kalamazoo, MI, USA) as a carrier and concluded that an antibiotic dosing regimen of either vancomycin or an aminoglycoside over 3.6 g per bag of cement was a statistically significant risk factor for AKI, with a 2-fold increased risk of AKI. Their mean dose of antibiotics was 17 g per spacer of a combination of vancomycin and an aminoglycoside with or without amphotericin B. Serum toxicity levels for these antibiotics were not available and, despite the study’s incredible value, it did not determine which antibiotic or combination of antibiotics in the spacer contributed to the AKI.

There is a paucity of studies detailing a clinically safe high end dosing regimen specifically for tobramycin in CS, PMMA, or their combination and documenting the serum tobramycin levels as a result of these high dose carriers. This present study specifically targeted tobramycin toxicity and the safe dosing in CS, PMMA, or their combination, as directly determined by the postoperative serum tobramycin levels and postoperative serum creatinine levels.

Runner et al. [23] documented a case of renal failure in 2018 following the use of a tobramycin articulating cement spacer for a hip infection using two packs of cement with 3 g of vancomycin and 3.6 g of tobramycin per pack of cement. The patient had a preoperative creatinine of 1 mg/dL. Random serum vancomycin level was 50.2 μg/mL on postoperative day 2 and at postoperative day 8 the serum creatinine was 6.52 mg/dL. Serum tobramycin level remained persistently elevated over the course of 2 weeks at 1 mcg/mL despite hemodialysis, requiring explantation of the spacer at 3 weeks postoperatively. Creatinine decreased to 2.67 mg/dL postoperatively following exchange spacer using plain cement. Serum tobramycin levels were useful in this case to follow systemic toxicity and guide treatment with respect to clearing the nephrotoxic drugs via dialysis.

In 2012, Menge et al. published a series of 84 patients receiving tobramycin and vancomycin impregnated cement spacers for infected total knee arthroplasties [18]. The incidence of AKI was 17%. Greater than 4 g of vancomycin in the spacer was associated with an odds ratio (OR) of 5.97 for developing AKI. The incidence of AKI was also directly associated with the use of tobramycin in the spacers with an OR of 5.87 for patients with >4.8 g of tobramycin when compared to those patients with spacers at doses < 4.8 g tobramycin. The authors then analyzed the tobramycin dose as a linear variable for developing AKI and the OR was 1.24 for every 1 g increase in the spacer tobramycin dose. Intriguingly, this association was not seen for each vancomycin g increased in the spacer. All patients with AKI had spacers containing tobramycin. They also found that intraoperative hypotension as well as the use of nephrotoxic medications was not statistically significantly associated with AKI. This was also the case in our study. In the report by Menge et al., 62% of their population received vancomycin intravenously postoperatively and the mean creatinine in the AKI population peaked at 30 days and remained elevated at 90 days postoperatively [18]. In the present study, our 7 patients with AKI peaked and resolved within 1 week postoperatively except for 1 patient who developed CKI that took 8 months to resolve.

Menge et al. recorded no postoperative serum tobramycin or vancomycin levels and recommended that “larger studies incorporating postoperative serum tobramycin monitoring are needed to determine the relation of tobramycin dosing to AKI” [18]. Our study does this by directly correlating the local tobramycin dose in the CS, PMMA, or both with the serum tobramycin levels. The rate of AKI in our study was low at 12% and our sole patient who went on to CKI received 8.4 g of tobramycin. The serum tobramycin levels in this patient were 1.41, 0.82, and 0.78 mcg/mL, respectively, at 24, 48, and 72 h. The mean tobramycin dose in our study was 9.7 g in the combination group. All three patients with over 12 g of tobramycin had high 24 h serum tobramycin levels and the patient receiving 19 g went on to AKI that did resolve.

This study has limitations, including its retrospective nature. The majority of cases in this study did not have preoperative underlying chronic kidney disease, with an incidence of only 22% (13/58), which would have adversely affected their ability to clear the systemic tobramycin. Hence, our study cannot determine directly what impact these local tobramycin dosing regimens may have on that population. Surgeons should proceed with caution when locally dosing antibiotics in these patients, as there have been reports of AKI in this population [21]. Additionally, biomarkers for declining kidney function were not used in our study. These include apelin, copeptin, and neutrophil gelatinase-associated lipocalin. These markers may be useful in future studies for preoperative as well as postoperative early indications of decline in kidney function [24,25].

The use of postoperative drains in the majority of the patients also may have had a diminishing effect on the initial amount of local antibiotic delivery, as shown by Vrabec et al. [26]. In their study, the medial intraarticular tobramycin level measured from hemovac samples at 6, 24, and 48 h decreased from 31.8, 17.1, and 6.8 μg/mL, respectively. In our study, vancomycin was used consistently in tandem with tobramycin in all cases, excluding those patients with vancomycin allergy. Vancomycin also is a contributor to kidney injury and has a synergistic bactericidal effect when used in tandem with tobramycin [18]. The doses in all cases of vancomycin were consistent at 1 g per 40 g of cement and 1 or 2 g per 20 mL of CS. The maximum vancomycin dose in all these cases was 9 g. This is well below concentrations reported in the literature and compared to the large doses of tobramycin used in our study, we do not think this adversely impacted our ability to evaluate the serum toxicity and blood levels of tobramycin and their effect on AKI.

## 4. Materials and Methods

After obtaining Institutional Review Board (IRB) approval, a retrospective review was performed on a consecutive series of patients from January 2016 to December 2018 at a single institution. A total of 52 patients were identified who were treated with tobramycin only or tobramycin/vancomycin antibiotic laden spacers (either CS, PMMA, or both). A total of 58 procedures occurred for the patient group for treatment of bone and soft tissue infection, or, in 3 cases, for prophylactic high-risk infection prevention. Inclusion criteria were as follows: (1) patients treated with tobramycin antibiotic laden spacers using CS and/or PMMA and (2) patients who had sufficient lab results quantifying the levels of tobramycin after implantation. Patients who had received systemic tobramycin were excluded in order to verify that the serum aminoglycoside levels obtained were related only to the local tobramycin laden spacer.

Antibiotic carriers used were either CS, PMMA, or both; the choice was individualized per case according to whether mechanical stability was required or not, with the decision made by the senior author. Patients in 10 procedures (17%) received only CS, patients in 10 procedures (17%) received only PMMA, and 38 procedures (66%) used a combination of PMMA and CS. For instances of PMMA spacers and implant coating, Palacos bone cement (Zimmer Biomet, Warsaw, IN, USA) was used, each pack of which contained 40 g of PMMA with 1 g of vancomycin and 3.6 g of tobramycin added to each pack, according to the recipe. Cement spacers were made using between 2 and 8 packs of antibiotic cement and implant coatings were made using 2 packs. The exact amount that was used depended on patient, implant size, and degree of bone; specifics follow in Results. CS was used for local antibiotic delivery for bone, soft tissue, and prosthetic joint infections requiring surgical debridement. Implantations ranged from 10 to 40 mL. When spacer or bead format was employed, 1.2 g of tobramycin and 2 g of vancomycin were added to 20 mL of CS. This was decreased to 0.6 g of tobramycin and 1 g of vancomycin when used as an injection to improve flow characteristics and delivery using a cement gun technique. Increased amounts of antibiotic were requested for 30–40 mL of CS (2.4 g of tobramycin/3 g of vancomycin, and 3.6 g of tobramycin/4 g of vancomycin, respectively).

Postoperatively, tobramycin blood samples were obtained at 24, 48, and 72 h. If a tourniquet was used during the surgery, its release time was considered as the implantation time. Blood was sampled in BD Vacutainer SST II Advance (Becton Dickinson, Franklin Lakes, NJ, USA) serum separator tubes. All samples were processed in the same manner by the same lab. Systemic toxicity levels of tobramycin in the literature are at levels > 12 mcg/mL [27], however, a critical tobramycin toxicity level at the authors’ institution was >8 mcg/mL and was based upon the institutional pharmacy and clinical lab reference ranges.

Patient age, sex, race, body mass index (BMI), weight, surgical location (involved body part and laterality), and relevant comorbidities were recorded. The amount of tobramycin delivered and the method of delivery (CS, PMMA, or both) were noted. Baseline and postoperative tobramycin and creatinine levels were measured with venous blood samples and cataloged. Normal low creatine in mg/dL was 0.5, normal high was 1.30, and critical high was >4.00. Nephrotoxic preoperative medications were documented and defined as the following: lisinoprol, metropolol, losartan, non-steroidal anti-inflammatory (NSAIDs), atorvastatin, hydrochlorothiazide, vancomycin, pravastatin, acetaminophen, losartan, or methotrexate.

Patients with a history of chronic kidney disease were recorded based on history and estimated glomerular filtration rate (GFR) of <60 mL/min. This was determined with the Chronic Kidney Disease Epidemiology Collaboration (CKD-EPI) equation, which requires age, sex, and race to calculate. Intraoperative intravenous antibiotic delivery was also noted in addition to any intraoperative hypotension that could inadvertently contribute to an AKI. An AKI was defined as an increase in creatine level of ≥0.3 mg/dL within any 48 h period post-implantation, or increase ≥ 1.5 times in accordance with the Kidney Disease: Improving Global Outcomes (KDIGO) guidelines [28]. Data was also collected regarding eradication of infection with follow-up reported at 1 year. Infection was determined to be eradicated at 1 year based upon the clinical examination and return of serologic markers (C-reactive protein (CRP) and erythrocyte sedimentation rate (ESR)) to normal in the absence of systemic or oral antibiotic administration.

After removing patient identifiers, the data was placed into Excel (Microsoft, Redmond, WA, USA) for initial formulation; subsequent statistical analysis was performed using MedCalc statistical software (version 20.009; MedCalc Software Ltd., Ostend, Belgium). Baseline demographics were summarized using mean with standard deviation (SD) and percentages of total procedural population. Descriptive statistics were calculated for continuous and categorical data from this study. Correlation analysis was performed to determine an association between tobramycin blood levels and tobramycin local antibiotics administered; 95% confidence interval (CI) for the Pearson correlation coefficient was reported with *p* value (significance was considered if <0.05). CI is reported between −1 and 1, with 1 portraying a positive correlation. This data was portrayed with a scatter diagram with a local regression smoothing trendline plotted.

## 5. Conclusions

To our knowledge, the present work is the largest series studying tobramycin blood levels and AKI with the use of local antibiotic tobramycin carriers in soft tissue and bone infection. This study suggests that a local tobramycin dosing < 9 g mixed with either CS, PMMA, or both results in safe postoperative tobramycin blood levels. The effect of the tobramycin blood levels on the incidence of AKI needs to be studied further.

It is the authors’ current practice to keep the tobramycin dose < 9 g in total when using CS, PMMA, or a combination thereof. Serum tobramycin levels are a useful adjunct in cases where there is a concern about renal toxicity. Nephrology should be consulted at the earliest sign of elevated tobramycin levels not decreasing < 2 mcg/mL at 72 h, especially in association with postoperative elevation in serum creatinine levels. All nephrotoxic intravenous antibiotics and medications should be stopped if possible. A low threshold for spacer exchange in addition to hemodialysis to decrease the tobramycin levels and the AKI is necessary to prevent chronic kidney injury [23].

In conclusion, local tobramycin antibiotic delivery using CS, PMMA or both remains a safe and effective modality in the treatment of osteomyelitis as long as the surgeon is aware of its possible nephrotoxic effect.

## Figures and Tables

**Figure 1 antibiotics-11-00336-f001:**
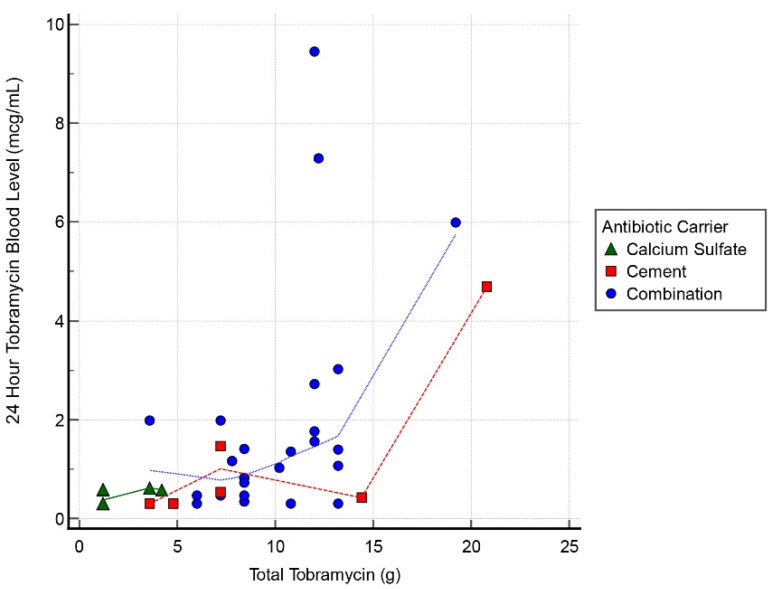
Scatter graph of tobramycin blood level (mcg/mL) and total tobramycin (g) at 24 h interval.

**Figure 2 antibiotics-11-00336-f002:**
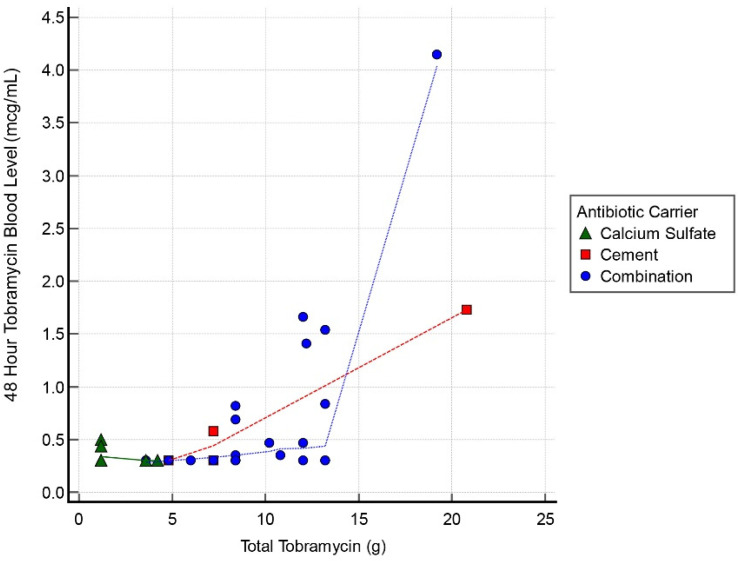
Scatter graph of tobramycin blood level (mcg/mL) and total tobramycin (g) at 48 h interval.

**Figure 3 antibiotics-11-00336-f003:**
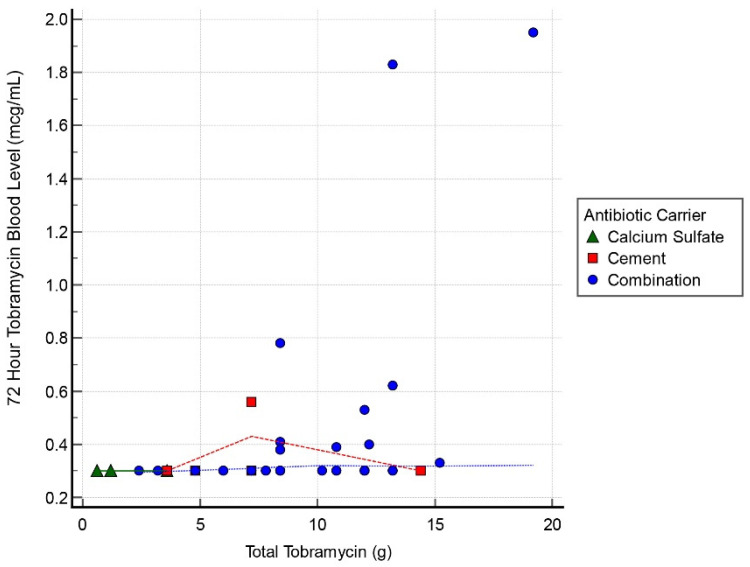
Scatter graph of tobramycin blood level (mcg/mL) and total tobramycin (g) at 72 h interval.

**Table 1 antibiotics-11-00336-t001:** Summary of baseline demographics at time of procedure.

Mean ± SD or (N (%))	Total Procedures (*n* = 58)
Age	61.7 ± 12.0 years
BMI	33.2 ± 8.8 kg/m^2^
Weight	96.9 ± 28.8
Male gender N (%)	36 (62.1%)
Comorbidities	
Atrial fibrillation	15 (25.9%)
Hypertension	44 (75.9%)
Diabetes mellitus	23 (39.7%)
Infectious disease	0 (0%)
Chronic kidney disease	8 (13.8%)
Surgical indication	
Periprosthetic knee infection	33 (56.9%)
Periprosthetic hip infection	12 (20.7%)
Osteomyelitis of tibia or femur	9 (13.8%)
TKA prophylaxis	2 (3.4%)
Prophylaxis in HTO excision	1 (1.7%)
Lumbosacral infected wound	1 (1.7%)
Baseline serum creatinine (mg/dL)	1.0 ± 1.0
Preoperative nephrotoxic medications	39 (67%)

BMI, body mass index; HTO, high tibial osteotomy; SD, standard deviation; TKA, total knee arthroplasty.

**Table 2 antibiotics-11-00336-t002:** Systemic and local antibiotic details.

	Dose	Total Procedures (*n* = 58)
Antibiotic Carrier		
CS only	14.0 ± 5.2 mL	10 (17.2%)
PMMA only	4.8 ± 2.9 units	10 (17.2%)
Combination (CS and PMMA)	CS: 15.8 ± 6.4 mL PMMA: 4.6 ± 2.0 units	38 (65.5%)
Local Antibiotics Utilized		
Tobramycin (g)	7.3 ± 5.0 g	
Vancomycin (g)	4.4 ± 2.4 g	
Systemic Antibiotics		
Cefazolin		31 (53.4%)
Clindamycin		9 (15.5%)
Vancomycin		8 (13.7%)
Ciprofloxacin		1 (1.7%)
Cefepime		1 (1.7%)
Unasyn		1 (1.7%)
Piperacilin and Tazobactam		1 (1.7%)
Unknown		9 (15.5%)
Two antibiotic combination		4 (6.9%)
Unasyn		1 (1.7%)
Piperacilin and Tazobactam		1 (1.7%)
Unknown		9 (15.5%)
Two antibiotic combination		4 (6.9%)

CS, calcium sulfate; PMMA, polymethyl methacrylate.

**Table 3 antibiotics-11-00336-t003:** Systemic tobramycin blood levels.

	Total Procedures (*n* = 58)
Detectable tobramycin blood levels (mean ± SD)	
24 h mcg/mL (*n* = 31)	1.87 ± 1.87
48 h mcg/mL (*n* = 17)	0.98 ± 0.96
72 h mcg/mL (*n* = 12)	0.71 ± 0.57
Undetectable levels (<0.3 mcg/mL) (N (%))	
24 h	11 (18.9%)
48 h	18 (31.0%)
72 h	43 (74.1%)
Tobramycin blood levels > 5.0 mcg/mL (N (%))	
24 h	3 (5.2%)
48 h	0 (0%)
72 h	0 (0%)
Correlation total tobramycin and blood levels	
24 h correlation coefficient (CI)	0.55 (*p =* 0.0002, 95% CI 0.29 to 0.73)
48 h correlation coefficient (CI)	0.63 (*p =* 0.0001, 95% CI 0.37 to 0.80)
72 h correlation coefficient (CI)	0.44 (*p =* 0.0016, 95% CI 0.18 to 0.65)

CI, confidence interval; SD, standard deviation.

**Table 4 antibiotics-11-00336-t004:** Acute kidney injury.

	Total Procedures (*n* = 58)
Stage 1	7 (12.0%)
Increase of 0.3 mg/dL within 48 h period	5 (8.6%)
≥1.5 times the baseline	2 (3.4%)
Stage 2	0 (0%)
Stage 3	0 (0%)

**Table 5 antibiotics-11-00336-t005:** Acute kidney injury cases.

	Age, Sex	BMI	CKI	Comorbidities	Medications	Indication for Surgery	Total Antibiotic Tobramycin	Cement Units	CS Amount (cc)
1	58, F	29.8	N	HTN	Losartan, metoprolol	TKA PJI	7.20	6	20 cc
2	74, M	33.6	Y	HTN, AF, DM	Vancomycin	TKA PJI	13.20	5	10 cc
3	55, M	34.9	N	HTN	Atrovastatin	TKA PJI	8.40	8	20 cc
4	52, M	30.9	N	HTN, DM	Vancomycin, losartan	TKA PJI	7.20	3	20 cc
5	71, F	39.6	Y	HTN	Lisonopril	Tibia FRI with retained implant	2.40	2	10 cc
6	56, F	19.9	N	None	None	TKA PJI	19.20	8	10 cc
7	60, M	49.2	N	HTN, DM	NSAIDs, atorvastatin	TKA PJI	NA	3	none

F, female; M, male; BMI, body mass index; CKI, chronic kidney insufficiency; N, no; Y, yes; HTN, hypertension; AF, atrial fibrillation; DM, diabetes mellitus; NSAIDs, non-steroidal anti-inflammatory drugs; TKA PJI, total knee arthroplasty with periprosthetic joint infection; FRI, fracture related infection; NA, not available.

## Data Availability

The data presented in this study are available upon request from the corresponding author. The data are not publicly available due to privacy concerns with protected health information.

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
