# Peer review of "Tobramycin Blood Levels after Local Antibiotic Treatment of Bone and Soft Tissue Infection"

_antibiotics, 2022, doi:10.3390/antibiotics11030336_

Round 1

Reviewer 1 Report

The authors carried out an interesting and original research. However, even if they are orthopedic specialists, some internal insights are missing.
It would be useful to make some evaluations:
1. The GFR measurement method is missing, even if the authors speak of estimated GFR (eGFR), but do not specify which method they used (MDRD, CKD-EPI). It is necessary to specify in the section “Materials and Methods” which method was used (I hope not creatinine clearance).
2. It is not specified whether patients who developed AKI had contraction of urine output and, if so, for how long. Please add these data in the "Results" section and enter them in the statistical analysis.
3. The authors do not specify the drugs that patients took before surgery. It is interesting to know if some drugs (for example metformin, ace inhibitors, ARBs) have been suspended and for how long before surgery. Please add these data.
4. The authors do not refer to the existence of biomarkers of kidney damage in AKI. In the "Discussion" section, reference should be made to the most important ones (for example NGAL, Nephroceck, urinary enzymes), adding references in the bibliography:
to. Bolignano D et al Nephrology 2010, 15 (1), pp 23-26
b. Lacquaniti A. et al Peptides 2013, 49, pp1-8

Author Response

  1. The GFR measurement method is missing, even if the authors speak of estimated GFR (eGFR), but do not specify which method they used (MDRD, CKD-EPI). It is necessary to specify in the section “Materials and Methods” which method was used (I hope not creatinine clearance).

Response: Lifebridge Laboratory uses the CKD-EPI equation to estimate GFR in adults, which requires age, sex, and race to calculate. Added at line 273.

2. It is not specified whether patients who developed AKI had contraction of urine output and, if so, for how long. Please add these data in the "Results" section and enter them in the statistical analysis.

Response: Unfortunately, we are not able to retrieve this from the medical records.

3. The authors do not specify the drugs that patients took before surgery. It is interesting to know if some drugs (for example metformin, ace inhibitors, ARBs) have been suspended and for how long before surgery. Please add these data.

Response: Most commonly used nephrotoxic medications, added at line 87: Out of the 39 patients who were on nephrotoxic medications preoperatively, 41% (n=16) were on statins, 30.8% (n=12) were on angiotensin-converting enzyme (ACE) inhibitors, 20.5% (n=8) were on vancomycin, and 17.9% (n=7) were on angiotensin II receptor blockers (ARBs).

  1. The authors do not refer to the existence of biomarkers of kidney damage in AKI. In the "Discussion" section, reference should be made to the most important ones (for example NGAL, Nephroceck, urinary enzymes), adding references in the bibliography:
    to. Bolignano D et al Nephrology 2010, 15 (1), pp 23-26
    b. Lacquaniti A. et al Peptides 2013, 49, pp1-8

Response: Added at line 227 (limitations): Also, biomarkers for declining kidney function were not used in our study. These include apelin, copeptin, and neutrophil gelatinase-associated lipocalin. These markers may be useful in future studies for preoperative as well as postoperative early indications of decline in kidney function. New refs added there too as was suggested, thank you.

Reviewer 2 Report

In the presented study where a retrospective review of consecutive series of patients from a single Institution were done, the authors examined the systemic toxicity associated with local application of tobramycin-impregnated PMMA, calcium sulfate alone or their combination as carriers. Authors aimed to assess the postoperative systemic tobramycin concentrations at three predetermined time points. Also, pre and postoperative renal functions were assessed for possible acute kidney injury as a result of local tobramycin application. A total of 52 patients who underwent 58 procedures with tobramycin-vancomycin loaded carriers were included in the study. At 72 hours after local implantation, all systemic tobramycin levels were found to be <2 mcg/mL. A total of 7 (12%) cases were reported with AKI. Authors concluded that local tobramycin dose of <9 g is safe and is associated with a low incidence of AKI.

The study results highlight the need for clinicians to optimize the dose of local tobramycin to <9 g. Overall, this is an interesting, clinically important and well written study.

However, I have few comments or suggestions for the authors to consider:

  1. Page 2, line 52-54 (Introduction): Provide reference.
  2. Page 2, line 69-70 (Introduction): Cite appropriate reference.
  3. Results: Please make the tables (Table 1 &2) readable by aligning the variables properly. In all tables, provide “n” first and then “%” in brackets. May be some of the results which are already given in the table could be removed from the text.
  4. Page 3, line 96-101 (Results): Is it case or procedure?
  5. Page 7, line 137 (Results): May be it would be nice to provide a table detailing the 7 AKI patients including the amount of carriers and local antibiotic used, what was the choice of pre or post-operative systemic antibiotics, co-morbid conditions etc.
  6. Page 8, line 246-48 (M&M): Cement 246 spacers were ……using 2 packs. Is this mean corresponding amount of antibiotics were also added to the spacers?
  7. Page 8, line 257-58 (M&M): Rephrase the sentence.
  8. Page 8 267-68 (M&M): Mention the methodology used for measurement of pre and postoperative tobramycin and creatinine levels.
  9. Conclusions: Please shorten the conclusion part.

Author Response

  1. Page 2, line 52-54 (Introduction): Provide reference.

Response: added new ref 11 here and in reference list: van Vugt TAG, Arts JJ, Geurts JAP. Antibiotic-Loaded Polymethylmethacrylate Beads and Spacers in Treatment of Orthopedic Infections and the Role of Biofilm Formation. Front Microbiol. 2019;10:1626. Published 2019 Jul 25. doi:10.3389/fmicb.2019.01626

  1. Page 2, line 69-70 (Introduction): Cite appropriate reference.

Response: Added citations for refs 18 and 21 here at line 71.

  1. Results: Please make the tables (Table 1 &2) readable by aligning the variables properly. In all tables, provide “n” first and then “%” in brackets. May be some of the results which are already given in the table could be removed from the text.

Response: Tables have been updated and pasted into the “revised-with-tracked-changes” manuscript.

  1. Page 3, line 96-101 (Results): Is it case or procedure?

Response:  cases (as stated)

  1. Page 7, line 137 (Results): May be it would be nice to provide a table detailing the 7 AKI patients including the amount of carriers and local antibiotic used, what was the choice of pre or post-operative systemic antibiotics, co-morbid conditions etc.

Response: new table added

Table 5. Acute Kidney Injury Cases.

Age, sex

BMI

CKI

Comorbidities

Medications

Indication for surgery

Total antibiotic Tobramycin

Cement units

CS amount (cc)

1

58, F

29.8

N

HTN

Losartan, metoprolol

TKA PJI

7.20

6

20 cc

2

74, M

33.6

Y

HTN, AF, DM

Vancomycin

TKA PJI

13.20

5

10 cc

3

55, M

34.9

N

HTN

Atrovastatin

TKA PJI

8.40

8

20 cc

4

52, M

30.9

N

HTN, DM

Vancomycin, Losartan

TKA PJI

7.20

3

20 cc

5

71, F

39.6

Y

HTN

Lisonopril

Tibia FRI with retained implant

2.40

2

10 cc

6

56, F

19.9

N

None

None

TKA PJI

19.20

8

10 cc

7

60, M

49.2

N

HTN, DM

NSAIDs, atorvastatin

TKA PJI

NA

3

none

F, female; m, male; BMI, body mass index; CKI, chronic kidney insufficiency; N, No; Y, yes; HTN, hypertension; AF, atrial fibrillation; DM, diabetes mellitus; NSAIDs, non-steroidal anti-inflammatory drugs; TKA PJI, total knee arthroplasty with periprosthetic joint infection; FRI, fracture related infection; NA, not available.

N.B: Perioperative antibiotics is not available for all patients

  1. Page 8, line 246-48 (M&M): Cement 246 spacers were ……using 2 packs. Is this mean corresponding amount of antibiotics were also added to the spacers?

Response: We are not certain we understand this request the way it is presented. Yes, the antibiotics were added to the spacers according to the recipe of 1 gm vancomycin and 3.6 gm tobramycin per bag of cement. We clarified this at line 262.

  1. Page 8, line 257-58 (M&M): Rephrase the sentence.

Response: Removed the last two words of the sentence (“after implantation”) to clarify at line 273.

  1. Page 8 267-68 (M&M): Mention the methodology used for measurement of pre and postoperative tobramycin and creatinine levels.

Response: Venous blood samples taken to measure tobramycin and creatinine levels. Clarified at lines 284-286.

creatinine in mg/dL
Normal low= 0.50
Normal high= 1.30
Critical high >4.00

  1. Conclusions: Please shorten the conclusion part.

Response: At line 330, the conclusion was shortened to state: In conclusion, local tobramycin antibiotic delivery using CS, PMMA or both remains a safe and effective modality in the treatment of osteomyelitis as long as the surgeon is aware of its possible nephrotoxic effect.

This was also updated in the abstract.

Reviewer 3 Report

The application of local antibiotics in cement and bone filler carriers is commonly used to treat PJI due to the high concentrations over extended periods require to kill biofilms. However, as the authors point out there are concerns that local antibiotic administration may lead to systemic toxicity. Little is known regarding the systemic levels of antibiotics resulting from local administration thus this is a useful contribution to the literature.

Specific comments

1) Table 2 would be easier to read if the antibiotic type and dose were put into separate columns.

2) I found the discussion of dosing in the results section unclear with respect to the units – i.e. how much antibiotic in how much carrier. i.e. what is “dose” in terms of mg? For example in the sentence “Ten cases received only CS with a mean dose of 14.0 mL (range, 10-20 mL)” I presume this means that “dose” means either a 10CC or 20CC pack of CS. Rather than presenting as the mean it would be clearer to say X patients had 10CC packs and Y had 20CC packs (if I am interpreting correctly). Part of the confusion may come from the fact that Stimulan is normally referred to in units of CC rather than mLs. It would be useful to explain the units of PMMA and CS. This is better explained in the methods but could be made clearer in the results section.

2) In the figures the post-op time is called “interval” while the main text says “post implantation”, possibly “post-operative” would be more appropriate but it should be consistent.

3) For graphs 1,2 and 3 it would be useful to indicate by symbol or color which patients had CS, PMMA or both and which had vanc and tob or just tob.

4) The main conclusion is “This study suggests that tobramycin dosing <9 g is safe and has a low incidence 296 of AKI based upon the postoperative serum tobramycin levels and postoperative creati-297 nine levels”. Can the authors clarify is this <9g total per surgical site regardless of how it is distributed in a particular carrier or combination of carriers?

5) Since PMMA and beads are most commonly used for PJI it would be interesting to break out and present these data from the surgery types. This would add value to the paper but I do not see it as a requirement for the publication.

Author Response

1) Table 2 would be easier to read if the antibiotic type and dose were put into separate columns.

Response: Tables have been updated and pasted into the “revised-with-tracked-changes” manuscript.

2) I found the discussion of dosing in the results section unclear with respect to the units – i.e. how much antibiotic in how much carrier. i.e. what is “dose” in terms of mg? For example in the sentence “Ten cases received only CS with a mean dose of 14.0 mL (range, 10-20 mL)” I presume this means that “dose” means either a 10CC or 20CC pack of CS. Rather than presenting as the mean it would be clearer to say X patients had 10CC packs and Y had 20CC packs (if I am interpreting correctly). Part of the confusion may come from the fact that Stimulan is normally referred to in units of CC rather than mLs. It would be useful to explain the units of PMMA and CS. This is better explained in the methods but could be made clearer in the results section.

Response: At line 104, added sentence “Six patients had 10 cc packs and 4 had 20 cc packs.” At line 108, added sentence “Eighteen combination patients received 10 cc packs, 19 received 20 cc packs, and 1 received a 40 cc pack.”

3) In the figures the post-op time is called “interval” while the main text says “post implantation”, possibly “post-operative” would be more appropriate but it should be consistent.

Response: Updated text from “post implantation” to “postoperative” and lines 127 and 128.

4) For graphs 1,2 and 3 it would be useful to indicate by symbol or color which patients had CS, PMMA or both and which had vanc and tob or just tob.

Response: Updated figs pasted into the “revised-with-tracked-changes” manuscript.

5) The main conclusion is “This study suggests that tobramycin dosing <9 g is safe and has a low incidence 296 of AKI based upon the postoperative serum tobramycin levels and postoperative creati-297 nine levels”. Can the authors clarify is this <9g total per surgical site regardless of how it is distributed in a particular carrier or combination of carriers?

Response: Based on our data, <9g of Tobramycin resulted in 24 hour blood levels of <2 mcg/mL as detailed in the 24 hour scatter diagram. The cement and combination of cement with calcium sulfate in this cohort was utilized with large defects which requires a higher amount of antibiotics. Given the total population in this cohort and low number of AKIs (7), we were unable to perform a multi-variate regression analysis to determine the local tobramycin dosage that might trigger kidney injury.

At line 316, text has been updated to state: This study suggests that a local tobramycin dosing <9 g mixed with either CS, PMMA, or both results in safe postoperative tobramycin blood levels. The effect of the tobramycin blood levels on the incidence of AKI needs to be studied further.

Also updated in abstract at line 20 to state: In conclusion, local tobramycin antibiotic delivery using PMMA, CS, or bothremains a safe and effective modality in the treatment of osteomyelitis as long as the surgeon is aware of its possible nephrotoxic effect. 

6) Since PMMA and beads are most commonly used for PJI it would be interesting to break out and present these data from the surgery types. This would add value to the paper but I do not see it as a requirement for the publication.

Response: Not a requirement for the publication.

Round 2

Reviewer 1 Report

The corrections are effective. The paper is well written with excellent scientific impact.